# Over Half of Falls Were Associated with Psychotropic Medication Use in Four Nursing Homes in Japan: A Retrospective Cohort Study

**DOI:** 10.3390/ijerph19053123

**Published:** 2022-03-07

**Authors:** Nozomu Oya, Nobutaka Ayani, Akiko Kuwahara, Riki Kitaoka, Chie Omichi, Mio Sakuma, Takeshi Morimoto, Jin Narumoto

**Affiliations:** 1Department of Psychiatry, Graduate School of Medical Science, Kyoto Prefectural University of Medicine, Kyoto 602-8566, Japan; n-oya@koto.kpu-m.ac.jp (N.O.); a-ku1128@koto.kpu-m.ac.jp (A.K.); m06032rk@koto.kpu-m.ac.jp (R.K.); comichi@koto.kpu-m.ac.jp (C.O.); jnaru@koto.kpu-m.ac.jp (J.N.); 2Department of Psychiatry, Maizuru Medical Center, Kyoto 625-8502, Japan; 3Department of Psychiatry, Shiga University of Medical Science, Otsu 520-2192, Japan; 4Department of Clinical Epidemiology, Hyogo College of Medicine, Nishinomiya 663-8501, Japan; mio@hyo-med.ac.jp (M.S.); morimoto@hyo-med.ac.jp (T.M.)

**Keywords:** fall, adverse drug event, nursing home, psychotropic drugs, polypharmacy, dementia

## Abstract

Medication use can increase the risk of falls and injuries in nursing homes, creating a significant risk for residents. We performed a retrospective cohort study over one year to identify the incidence of drug-related falls with and without injury among four Japanese nursing homes with 280 beds. We evaluated the relationship between potential risk factors for falls and fall-related injuries while considering well-known risks such as ADLs and chronic comorbidities. By collaboratively reviewing care records, we enrolled 459 residents (mean age, 87) and identified 645 falls, including 146 injurious falls and 16 severe injurious falls requiring inpatient care, incidence: 19.5, 4.4, 0.5 per 100 resident-months, respectively. Medication influenced around three-quarters of all falls, >80% of which were psychotropic drugs. Regularly taking ≥5 medications was a risk factor for the initial falls (HR 1.33: CI 1.00–1.77, *p* = 0.0048) and injuries after falls (OR 2.41: CI 1.30–4.50, *p* = 0.006). Our findings on the incidence of falls with and without injury were similar to those in Western countries, where the use of psychotropic medication influenced >50% of falls. Discontinuing unnecessary medication use while simultaneously assessing patient ADLs and comorbidities with physicians and pharmacists may help to avoid falls in nursing homes.

## 1. Introduction

Aging is a global trend, especially in developed countries, and the number of older adults receiving care services is also growing. Japan has the highest aging population globally, with 28.1% of the population over the age of 65 years [1]. The number of older adults who require care services in Japan has increased from 1.8 million in 2000 to 5.5 million in 2018 [2]. As a result, the number of older adults living in elderly care facilities, such as nursing homes, has increased to 2.1 million in 2018 [3]. The situation is similar in the United States where 1.9 million adults of 65 years and over have received some care services in elderly care facilities [4], accounting for 12% of the older adults and 33% of all recipients of elderly care services in the United States. This suggests that many developed countries will face an increase in the number of elderly residents in the future. 

Falls and fall-related injuries are leading problems in nursing homes and other elderly care facilities. Approximately 50% of individuals in nursing homes fall each year [5], compared to 20–33% of the general elderly population who experience falls annually [6]. On occasion, falls can lead to injuries, and an estimated 2.6–25% of all falls in nursing homes caused significant injuries including hip fractures or cerebral hemorrhages [5,7,8,9]. The incidence of falls in nursing homes worldwide was reported to be 11.4 per 100 resident-months in Germany [10], 19.4 per 100 resident-months in the United States [11], and 20 per 100 resident-months in Spain [12]. In Japan, the incidence of falls in long-term care wards in a general hospital was reported to be 11.4 per 100 patient-months [13]; however, the incidence in nursing homes has not been reported. 

Many risk factors for falls have been reported in nursing homes or other clinical settings, for example, advanced age, history of falls [13], experiencing dementia [11], environmental hazards [14], using psychotropic medications [15,16], and consuming a large quantity of drugs [17,18]. Although some of these factors are unpreventable, the influence of medication could be reduced by improving the quality of prescription criteria according to proper prescribing guidelines for the elderly, such as Beers criteria. In addition, the proportion of falls in elderly care facilities that are influenced by medication use is still unclear. Furthermore, we did not find any previous retrospective or prospective studies that surveyed the relationship between falls and medication use by studying resident care records and all prescriptions in nursing homes. Thus, we aimed to identify the incidence and nature of falls with and without injury in four nursing homes in Japan, the most rapidly aging society in the world. In addition, we also identified the proportion of falls influenced by medication use and assessed the relationship between potential risk factors for falls and injuries after falls.

## 2. Materials and Methods

### 2.1. Study Design and Resident Population

We conducted a retrospective cohort study in four nursing homes in Japan with 280 beds. We included all residents except for short-term admissions from 1 August 2016 to 31 July 2017, regardless of whether the resident was admitted or discharged during the study period. Short-term residents were excluded because they usually resided in their home and were only admitted to the facilities for a few days per week, mainly for family respite purposes, and their characteristics were quite different from those of other residents. 

The Institutional Review Board of the Kyoto Prefectural University of Medicine approved this study. This study was performed according to the regulations set forth in the Declaration of Helsinki [19]. 

### 2.2. Definition

The primary outcome of this study was the number of falls, defined as unintentionally coming to the ground or some lower level for a reason other than as a consequence of sustaining a violent blow, loss of consciousness, sudden onset of paralysis in stroke, or an epileptic seizure [20]. For all falls, the presence of injury and the potential triggers of falls including medications, physical conditions, psychiatric conditions, operation errors of walking aids, slippage of the feet, trouble with clothes and shoes, obstacles, and mistakes during caregiving were assessed. In particular, if the medication was considered to be a potential trigger for the fall, the fall was defined as an adverse drug event (ADE) [21]. Fall injury was defined as a body lesion at the organic level, resulting from acute exposure to energy in amounts that exceed the threshold of physiological tolerance [22]. The injuries were classified as bruises, scratch wounds, subcutaneous hematomas, contusions, fractures, and intracranial hemorrhages. Falls with no apparent injury were classified as falls without injury. When residents were taken to the hospital immediately after the fall and left their nursing home on the same day, they were considered to have a “need for admission”.

Drugs were broadly classified into psychotropic and non-psychotropic drugs. Non-psychotropic drugs included antibiotics, antihypertensives, cardiovascular drugs, anticoagulants, antihyperlipidemic drugs, antidiabetics, peptic ulcer drugs, laxatives, anti-inflammatory drugs, corticosteroids, antiallergic agents, electrolytes or fluids, and Chinese herbal medicines. Psychotropic drugs included antipsychotics (atypical), antipsychotics (typical), antidepressants (SSRI, SNRI, NaSSA), antidepressants (other), sedatives (benzodiazepines and non-benzodiazepines), sedatives (other), anxiolytics, antiepileptics, anti-parkinsonian drugs, and antidementia medicines. We considered all benzodiazepines and non-benzodiazepines to be sedatives and anxiolytics as benzodiazepine receptor agonists (BZDRAs). In addition, we defined polypharmacy as “the use of five or more medications”, as per the common consensus [23]. 

### 2.3. Data Collection Process

The review process in this study was divided into three steps (Figure 1). First, five physicians and six clinical psychologists collaborated regarding the collection of data, such as sex, age, medical history and comorbidities, presence of dementia and functional independence at the beginning of this study, and all prescriptions, based on resident care records, prescriptions, laboratory results, and incident reports during the study period. Each fall record was written daily on resident care records and incident reports upon observation by the nurses and staff in the nursing homes. Comorbidities were quantified using the Charlson Comorbidity Index (CCI) score [24]. The Barthel Index (BI) was used for the evaluation of the functional independence of activities of daily life (ADLs) [25]. CCI and BI were calculated using the information from each care record. At the same time, care records were reviewed and suspected-fall descriptions were collected. For example, residents had slipped, stumbled, fell onto their knees, crouched, or were found on the floor.

Second, after collecting information on the suspected events, all physicians independently reviewed whether each suspected event adhered to the definition of fall and assessed the potential trigger of the fall, presence of injuries, and the treatment required, including transport to the hospital. Descriptions of residents that fell after conflict with other residents were excluded. When the following situations were applicable, physicians evaluated whether each fall was an ADE and which medication could have influenced the fall the most, using the Naranjo algorithm [26]. Namely, whether medication would cause a fall due to its pharmacological action, whether medication was administered or discontinued within 24 h of the occurrence of fall, and if the administration or discontinuation of the medication could be the potential trigger for the fall. For the assessment of ADEs, the reviewers were trained according to a review manual based on the previous methods [27].

Finally, physicians reviewed the medical records again and collaboratively confirmed the final classification for each fall and injury. When they failed to agree on how to classify a fall, the discussion was continued until a consensus was reached. The degree of agreement between evaluators was calculated using Cohen’s kappa.

### 2.4. Data Analysis

We calculated incidences per 100 resident-months and 95% confidence intervals (95% CIs). We compared characteristics between residents who fell during the study period and those who did not. Furthermore, we compared characteristics between residents who were injured after falls and those who were not using *t*-tests and Wilcoxon rank sum tests for continuous variables and the χ^2^ test for nominal variables. We present continuous variables as means with standard deviations (SDs) or medians with interquartile ranges (IQRs) and categorical variables as numbers and percentages.

Multivariate Cox proportional hazards models were used to compare the association of potential risk factors of first falls as the occurrence of falls, adjusting for differences in the length of hospital stay during the study period. In addition, we also assessed the relationship between potential risk factors and the occurrence of injurious falls using multivariate logistic regression models among the residents who experienced at least one fall during the study period. The potential risk factors in both adjusted models, the multivariate Cox proportional hazards model, and the multivariate logistic regression model included sex, age (≥85 years), presence of dementia, number of medications at the beginning of this study, CCI (≥3), and BI (independent and mildly dependent (≥60)). All statistical analyses were conducted using statistical analysis software JMP version 14 (SAS Institute, Cary, NC, USA), and two-sided *p*-values less than 0.05 were considered statistically significant.

## 3. Results

### 3.1. Characteristics of the Study Residents and the Incidence of Falls

A total of 459 residents were enrolled during the study period (Table 1). The mean age of the residents was 87 years (SD: 6.9), and about three-quarters of them were female. Almost half of residents were prescribed five or more medications at the beginning of the study period, and 88% had dementia. Regarding the number of medications, there was no difference in the degree of independence between residents in the non-faller category (*p* = 0.27). However, between those in the faller group, the number of medications was significantly higher in residents of low ADLs (BI < 60, *p* = 0.03).

During the study period, 645 falls occurred in 204 residents (44.4%) (Figure 2). The incidence of all falls was 19.5 per 100 resident-months (95% CI: 18.0–21.0). About a quarter of falls (146/645) in 89 residents led to some injuries, and 3.5% (16/459) of residents required transfer to hospital for admission due to severe injuries (bone fracture and intracranial hemorrhage). Therefore, the incidence of injurious falls and severe injurious falls requiring inpatient care was 4.4 (95% CI: 3.7–5.1) and 0.5 (95% CI: 0.3–0.7) per 100 resident-months, respectively. We identified potential triggers of falls (Table 2), and as a result, approximately three-quarters of falls (74.4%, 480/645) were influenced by medication immediately before the fall (i.e., ADEs), with psychotropic drugs accounting for 86.8% of all medications. Among these drugs, BZDRAs were the most common drugs that potentially triggered falls (37.3%, 179/480), followed by antipsychotics (21.7%, 104/480), antiepileptics (13.1%, 63/480), antihypertensives (9.8%, 47/480) antidementia drugs (5.4%, 26/480), and anti-Parkinson drugs (50%, 24/480). The Cohen’s kappa value was estimated using the results of an independent review by two physicians using a random sample of 60 of the 645 suspicious events obtained in the first part of the review. The kappa score between reviews for the occurrence of falls influenced by medication was 0.71 [95%CI 0.48–0.93].

### 3.2. Comparison of the Characteristics and Risk Factors between Fallers and Non-Fallers among All Residents

We compared fallers and non-fallers based on residents’ characteristics at the beginning of this study. We found that the fallers took more medications (median 5 vs. 4, *p* = 0.002) and more psychotropic drugs (median 1 vs. 0, *p* = 0.0008), especially BZDRAs (33% vs. 18%, *p* = 0.0003) and antiepileptics (19% vs. 11%, *p* = 0.001). Moreover, fallers exhibited a higher BI score (median 65 vs. 45, *p* = 0.0001) than non-fallers, even though fallers had fewer comorbidities than non-fallers (median 1 vs. 2, *p* = 0.006).

The Cox proportional hazard model estimation showed that polypharmacy was associated with a higher rate of fall occurrence at any time during follow-up (HR 1.33: CI 1.00–1.77, *p* = 0.048). Residents who were less dependent in ADLs also displayed a higher risk of falling (HR 1.44: CI 1.08–1.92, *p* = 0.01) (Table 3). Additionally, residents taking over ten medications also presented a high risk for falls according to the Cox proportional hazard model (HR 2.38, 95% CI 1.3–4.3).

### 3.3. Comparison of Characteristics and Risk Factors between Fallers with and without Injuries

The characteristics of fallers with injuries were similar to those of fallers without injuries except for functional independence in ADLs based on the BI score (Table 4). Logistic regression analyses among fallers showed that polypharmacy and less dependence in ADLs were associated with about a three-fold increase in the rate of fall injuries (OR 2.41: CI 1.30–4.50, *p* = 0.006, OR = 3.46: CI 1.84–6.54, *p* = 0.0001, respectively). The presence of dementia approached but did not reach statistical significance regarding an increase in the risk of fall injury (OR 2.52: CI 0.87–7.30, *p* = 0.09) (Table 5).

## 4. Discussion

We conducted a comprehensive epidemiological survey of falls in four nursing homes in Japan, including the incidence of all falls with and without injury and evaluated potential triggers for the occurrence of the falls and associations with the characteristics of the residents, such as polypharmacy, presence of dementia, and functional independence of ADLs. Since many residents in nursing homes experience dementia [28] and often take psychotropic drugs [29], psychiatrists and clinical psychologists as experts of behaviors and treatments of individuals with dementia took the lead in conducting this study under the supervision of two internists with sufficient experience in clinical epidemiological research.

The incidence of all falls in this study was 19.5 per 100 resident-months, similar to values in previous studies from Spain (20 per 100 resident-months), Germany (11.4 per 100 resident-months), and the United States (19.4 per 100 resident-months) [10,11,12]. In addition, the proportion of falls with injuries (23%, 146/645) and hip fractures after falls (2.3%, 15/645) were similar to those in nursing homes in the United States (15% and 2.6%, respectively) [5,30]. Although the methods of calculation among the studies were inconsistent and thus cannot be simply compared, the incidence of falls in our study was similar to that of previous studies.

Focusing on potential triggers for the occurrence of falls, we found that about three-quarters of falls were influenced by medication use, with psychotropic drugs accounting for more than 80% of all drugs. In other words, over half of all falls were influenced by the use of psychotropics in these nursing homes. Many psychotropic drugs are used to treat behavioral and psychological symptoms of dementia (BPSD) [18,31]. The avoidance of antipsychotics and BZDRA to the greatest extent possible has been recommended for older adults with dementia because of the risk of ADEs [32,33]. Furthermore, psychotropic drugs such as antipsychotics, benzodiazepines, and antidepressants increase the risk of falls in nursing home residents [16,32]. In fact, BZDRAs and antipsychotics were two of the most frequently suspected drugs for risk of falls in this study (37.3% and 21.7%, respectively), and the frequency of falls caused by antiepileptics was also found with high frequency in this study (13.1%, 63/480). The high number of falls caused by antiepileptics may be a result of commissioned physicians’ attempt to avoid the use of BZDRAs and antipsychotics, which have been shown to be harmful to older adults with dementia. Antiepileptics were sometimes used for the treatment of BPSD, such as agitation and aggression, with the expectation of mood-stabilizing effects despite not being strongly recommended in the guidelines for the treatment of dementia in Japan; however, physicians should be aware that valproate and other antiepileptics have insufficient evidence in the management of BPSD and that the ADEs of these drugs are sometimes problematic [34].

The results of this study empirically demonstrate the influence of psychotropic drugs on falls in four nursing homes. This result suggests that adequate assessment of the need for psychotropic drug use for treating BPSD and the appropriate monitoring of the risk of falls after prescribing the drug are particularly important in preventing falls in nursing homes. Therefore, doctors themselves should be careful about their prescribing and monitoring and nurses and carers should be educated on medications and interventional programs [35,36]. Moreover, psychotropics could have a greater effect on the frequency of falls in comparison to physical medications, as non-fallers had more comorbidities than fallers despite their higher number of comorbidities.

The study also showed that receiving polypharmacy and a high level of independence in ADLs were associated with a significantly higher risk for the occurrence of falls and injuries after falls. Some previous studies have shown that polypharmacy is less related to decreasing ADL levels [37], and there was no association between polypharmacy and ADL reduction in nursing home residents [38]. This implies that polypharmacy has little effect on a resident’s ADL level. Therefore, care providers need to be aware that polypharmacy in patients with high ADL levels may increase the risk of falls and fall-related injuries due to the dual risk factors of polypharmacy and high ADL levels.

There are some limitations to this study. First, this study was conducted at four facilities classed as nursing homes in a small region: this type of facility accounts for almost half of long-term care facilities for older adults in Japan, excluding residents in assisted living facilities and sanatoriums [2]. Further research is needed to better understand the epidemiology of falls among inhabitants of long-term care facilities for the elderly population in Japan as a whole. Second, because this study was conducted through a retrospective chart review of the care records, we were unable to assess the potential triggers of falls that were not recorded in the care records. In addition, the influence of medication on the occurrence of falls may have been overestimated because we were able to access the extreme detail of the prescriptions in the records. Finally, we could not assess the risk of falls and fall-related injuries for the use of specific medications statistically because we examined the residents’ medications at the beginning of the survey as characteristic information for the residents in this study. Therefore, in cases where drugs were thought to have influenced falls, we used established methodologies to evaluate the medication that had an influence on the occurrence of falls [27]. Multivariate analysis was used to assess the association of drugs as a risk factor for falls and fall-related injuries, but only included the number of medications taken at the beginning of the survey. In the strict sense, there were some differences in prescriptions between those at the beginning of survey and those at the time of falls. However, previous studies also used the factors at the beginning of the survey as representative values for each resident in the analysis [13,39]. Furthermore, residents in nursing homes generally do not have a dramatic change in prescriptions during their residency.

## 5. Conclusions

We assessed the incidence and nature of falls in four Japanese nursing homes, and the incidence of both falls and injurious falls in Japan were comparable to that in Western countries. About three-quarters of all falls were influenced by medication, and over half of all falls could be affected by psychotropic medication use such as BZDRAs, antipsychotics, and antiepileptics. We believe that adequate assessment of their comorbidities and medication and monitoring for psychotropic drugs are particularly important to prevent falls in nursing homes.

## Figures and Tables

**Figure 1 ijerph-19-03123-f001:**
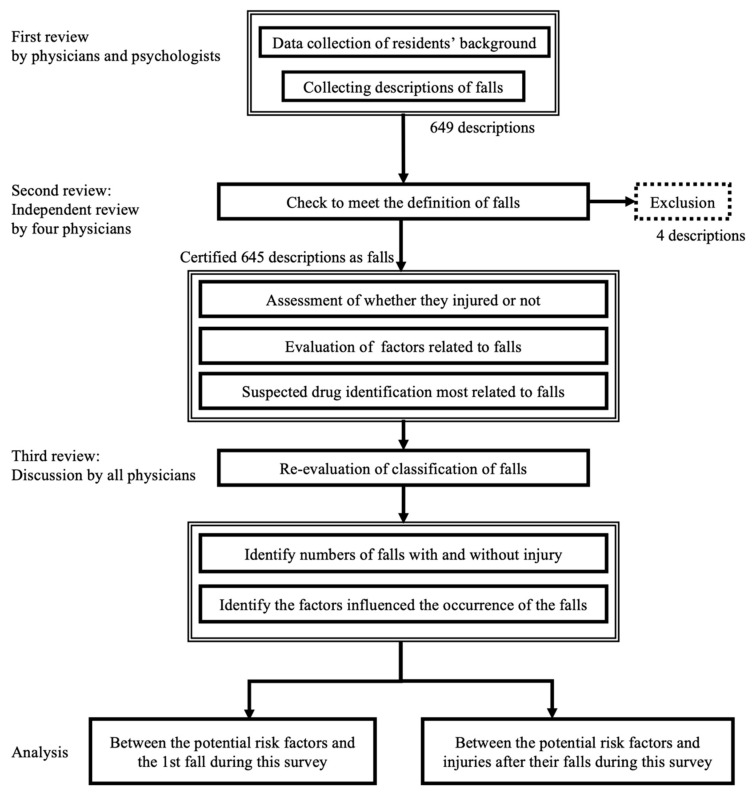
Outline of review process.

**Figure 2 ijerph-19-03123-f002:**
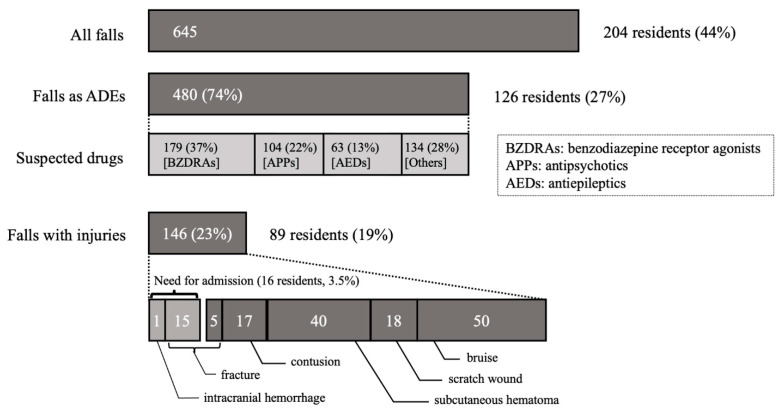
Severity of falls.

**Table 1 ijerph-19-03123-t001:** Characteristics of residents.

Variables	ALL (*n* = 459)	Fallers ^†^ (*n* = 204)	Non-Fallers (*n* = 255)	Χ^2^ or t Value	*p*-Value
Female, *n* (%)	344 (75)	157 (77)	187 (73)	0.79	0.4
Age, mean (SD) (years)	87.0 (6.9)	87.3 (6.5)	86.9 (7.2)	0.50	0.7
Length of stay, median (IQR) (days)	113 (0–760)	148 (0–776)	104 (0–738)	−0.48	0.6
No. of all drugs ^§^, median (IQR)	4 (2–6)	5 (3–7)	4 (2–6)	2.37	0.02
Polypharmacy ^§,¶^, *n* (%)	210 (46)	105 (51)	105 (41)	4.84	0.03
No. of psychotropic drugs ^§^, median (IQR)	0 (0–1)	1 (0–2)	0 (0–1)	3.55	<0.001
Psychotropic drugs ^§,#^, *n* (%)	228 (49.6)	115 (56)	113 (44)	6.61	0.01
BZDRAs ^§,$^, *n* (%)	115 (25)	68 (33)	47 (18)	13.4	<0.001
Antipsychotic drugs ^§^, *n* (%)	332 (72)	59 (29)	68 (27)	0.29	0.6
Antiepileptic drugs ^§^, *n* (%)	65 (14)	38 (19)	27 (11)	6.73	0.01
Antidepressant drugs ^§^, *n* (%)	29 (6.3)	16 (7.8)	13 (5.1)	1.43	0.2
Antihypertensive drugs ^§^, *n* (%)	206 (45)	95 (47)	111 (44)	0.42	0.5
Dementia, *n* (%)	406 (88)	181 (89)	225 (88)	0.03	0.9
Charlson Comorbidity Index, median (IQR)	1 (1–3)	1 (1–3)	2 (1–3)	−2.47	0.006
Body mass index, median (IQR)	19.7 (17.5–21.9)	20.3 (18.0–22.3)	19.0 (17.1–21.8)	1.92	0.06
Barthel Index), median (IQR)	50 (25–80)	65 (45–80)	45 (10–75)	5.83	<0.001
Barthel Index ≥ 60 (independent, mildly dependent), *n* (%)	213 (46)	114 (56)	99 (39)	13.3	<0.001

SD, standard deviation; IQR, interquartile range; BZDRAs, benzodiazepine receptors agonists (benzodiazepines and non-benzodiazepines). ^†^ Residents who fell at least once during this study. ^§^ Presence of prescriptions at the beginning of this study including antipsychotics, antidepressants. ^¶^ The number of medications was five or more at the beginning of this study. ^#^ Including antipsychotics (atypical), antipsychotics (typical), antidepressants (SSRI, SNRI, NaSSA), antidepressants (other), sedatives (benzodiazepines and non-benzodiazepines), sedatives (other), anxiolytics, antiepileptics, anti-parkinsonian drugs and antidementia medicines. ^$^ Including sedatives and anxiolytics as benzodiazepine receptors agonists.

**Table 2 ijerph-19-03123-t002:** Factors influencing each fall.

Factors (Multiple Selections Were Possible)	
No. of falls, *n*	645
Medications (i.e., ADEs *^1^), *n*(%)	480 (74)
Physical conditions *^2^, *n*(%)	375 (58)
Psychiatric conditions, *n*(%)	315 (49)
Operation errors of wheelchairs, *n*(%)	25 (3.9)
Mistakes during caregiving, *n*(%)	25 (3.9)
Slip of their feet, *n*(%)	12 (1.9)
Trouble with clothes(trousers)/shoes, *n*(%)	9 (1.4)
Obstacles at their feet, *n*(%)	8 (1.2)
Operation errors of walking frames, *n*(%)	7 (1.1)
Troubles with others *^3^, *n*(%)	4 (0.6)
Unspecified, *n*(%)	29 (4.5)

*^1^ ADE; adverse drug event; *^2^ Including impaired walking patterns, impaired balance, reduced muscle strength, acute medical illness; *^3^ Excluding falls without other causes.

**Table 3 ijerph-19-03123-t003:** Hazard ratios of fall occurrence from Cox proportional hazard models.

Risk Factor	Crude HR	95% CI	Adjusted HR	95% CI
No. of medication ^¶^ ≥ 5	1.37	1.04–1.80	1.33	1.00–1.77
Female	0.86	0.63–1.21	0.80	0.57–1.11
Age ≥ 85	0.96	0.72–1.28	1.02	0.75–1.37
Dementia	0.97	0.64–1.53	1.03	0.65–1.63
CCI ≥ 3	0.82	0.60–1.11	0.83	0.60–1.15
BMI < 20	0.81	0.61–1.07	0.89	0.67–1.18
Barthel Index ≥ 60 (independent, mildly dependent)	1.48	1.12–1.95	1.44	1.08–1.92

HR, hazard ratio; CI, confidence interval; CCI, Charlson comorbidity index; BMI, body mass index. ^¶^ The number of medications was five or more at the beginning of this study.

**Table 4 ijerph-19-03123-t004:** Characteristics between fallers with and without injuries.

Variables	All Fallers (*n* = 204)	Fallers with Injuries (*n* = 89)	Fallers without Injuries (*n* = 115)	Χ^2^ or t Value	*p*-Value
Female, *n* (%)	157	68 (76)	89 (77)	0.03	0.9
Age, mean (SD) (years)	87.3	87.1 (6.6)	87.4 (6.4)	−0.39	0.7
Length of stay, median (IQR) (days)	148 (0–776)	169 (0–872)	120 (0–649)	1.26	0.2
No. of all drugs ^§^, median (IQR)	5 (3–7)	5 (3–6.5)	4 (2–7)	0.61	0.3
Polypharmacy ^§,¶^, *n* (%)	105	52 (58)	53 (46)	3.07	0.08
No. of psychotropic drugs ^§^, median (IQR)	1 (0–2)	1 (0–2)	1 (0–2)	0.12	0.8
Psychotropic drugs ^§^, *n* (%)	115	53 (60)	62 (54)	0.65	0.4
BZDRAs ^§^, *n* (%)	68	32 (36)	36 (31)	0.49	0.5
Antipsychotic drugs ^§^, *n* (%)	59	25 (28)	34 (30)	0.05	0.8
Antiepileptic drugs ^§^, *n* (%)	38	15 (17)	23 (20)	0.07	0.6
Antidepressant drugs ^§^, *n* (%)	16	4 (4)	12 (10)	2.59	0.1
Antihypertensive drugs ^§^, *n* (%)	95	43 (48)	52 (45)	0.19	0.7
Dementia, *n* (%)	181	82 (92)	99 (86)	1.89	0.2
Charlson Comorbidity Index, median (IQR)	1 (1–3)	1 (1–3)	1 (1–3)	0.46	0.6
Body mass index, median (IQR)	20.3 (18.0–22.3)	20.2 (18.1–22.0)	20.3 (17.8–22.5)	−0.21	0.8
Barthel Index, median (IQR)	65 (45–80)	70 (50–90)	50 (40–75)	4.30	<0.0001
Barthel Index ≥ 60 (independent, mildly dependent), *n* (%)	114	64 (72)	50 (43)	16.8	<0.0001

SD, standard deviation; IQR, interquartile range; BZDRAs, benzodiazepine receptors agonists (benzodiazepines and non-benzodiazepines); ^§^ Presence of prescriptions at the beginning of this study. ^¶^ The number of medications was five or more at the beginning of this study.

**Table 5 ijerph-19-03123-t005:** Odds ratios of injury occurrence after falls based on logistic regression analysis.

Risk Factor	Crude OR	95% CI	Adjusted OR	95% CI
No. of medication ^¶^ ≥ 5	1.64	0.94–2.87	**2.41**	**1.30–4.50**
Female	0.95	0.49–1.82	0.91	0.44–1.87
Age ≥ 85 y	0.91	0.51–1.63	0.94	0.49–1.77
Dementia	1.89	0.74–4.82	2.52	0.87–7.30
CCI ≥ 3	0.96	0.51–1.79	0.98	0.50–1.96
BMI < 20	1.10	0.63–1.94	1.29	0.70–2.37
Barthel Index ≥ 60 (independent, mildly dependent)	3.33	1.84–6.01	3.46	1.84–6.54

OR, odds ratio; CI, confidence interval; CCI, Charlson Comorbidity Index; BMI, body mass index. ^¶^ The number of medications was five or more at the beginning of this study.

## Data Availability

On reasonable request, derived data supporting the findings of this study are available from the corresponding author after approval from the Institutional Review Board of Kyoto Prefectural University.

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
