# Peer review of "Over Half of Falls Were Associated with Psychotropic Medication Use in Four Nursing Homes in Japan: A Retrospective Cohort Study"

_ijerph, 2022, doi:10.3390/ijerph19053123_

Round 1
Reviewer 1 Report
Thank you for the opportunity to review this paper. Measuring changes over time that reduce the risk of falls would be more clinically relevant and add value to the study. Below are some specific comments:
Abstract
- “…incidence: 19.5, 4.4, 0.5 per 100 resident-months, respectively)… This is unclear, i.e respective to what?
- The authors conclude: Continuous assessment to prevent polypharmacy and careful monitoring when prescribing psychotropic drugs are imperative in preventing falls in nursing homes.
- How do the authors propose to “prevent” polypharmacy in older individuals who are most likely to have multiple chronic morbidities?
Introduction
- Requires grammatical edits.
- Lines 55-56: “Although some of these factors are unpreventable, the influence of medication might be reduced by improving prescriptions.”
- Authors may want to reconsider the use of “improving prescriptions”. Do they mean current prescriptions are suboptimal?
- Lines 57-58: “In addition, it is still unclear worldwide what proportion of falls in elderly care facilities are influenced by medication use” This is not a global study and hence authors may need to rephrase this to make it more relevant to their study.
- The authors claim that this is a historical study however they are focusing on the incidence and nature of falls. Authors need to clearly justify that such as study has not been conducted before if they are to call it “historical”.
Methods
- Figure 1 – include the number of people in your flow chart if possible, e.g. data was collected for how many patients? How many were excluded?
- How did the authors directly associate a fall as an adverse event of a drug? This needs to be more elaborate.
Results
- The authors need to clearly identify and eliminate all other influencing factors that could have led to a fall apart from medicines. Once this is addressed in the methods, the results will be corroborated. Currently, the elimination of other factors is vague and therefore statements in results such as the proportion of people with medication-related falls and specific medicines need validation.
- “3.2. Comparison of the characteristics and risk factors between fallers and non-fallers among all 185 residents”. It would be interesting to discuss reasons why fallers had more medications, compared to non-fallers, despite having fewer morbidities.
Discussion
- Lines 240-246: “high number of falls caused by antiepileptics may be a result of commissioned physicians’ attempt to avoid BZDRA and antipsychotics, which have been shown to be harmful to older adults with dementia…”
- If authors are implicating that antiepileptics are used for other purposes (not for epilepsy), this should be discussed clearly and mentioned if this is according to standard practice guidelines in Japan.
Author Response
Point 1: • “…incidence: 19.5, 4.4, 0.5 per 100 resident-months, respectively)… This is unclear, i.e respective to what?
Response 1: Thank you for your comments. These mean falls, injurious falls, and severe injurious falls. We modified this sentence (page 1, line 21-23)
Point 2: • The authors conclude: Continuous assessment to prevent polypharmacy and careful monitoring when prescribing psychotropic drugs are imperative in preventing falls in nursing homes. How do the authors propose to “prevent” polypharmacy in older individuals who are most likely to have multiple chronic morbidities?
Response 2: Thank you for your comment. To make our message clearly, we have made the following corrections to the section you pointed out as below; Preventing the occurrence and persistence of polypharmacy by discontinuing unnecessary medications may lead avoid falls in nursing homes while consulting with physicians and pharmacists after adequate assessment of ADLs and comorbidities (page 1, line 28-30).
Point 3: • Lines 55-56: “Although some of these factors are unpreventable, the influence of medication might be reduced by improving prescriptions.” Authors may want to reconsider the use of “improving prescriptions”. Do they mean current prescriptions are suboptimal?
Response 3: Thank you for your suggestion, In Japanese clinical setting in nursing homes, residents in nursing homes often take medications against the propre prescribing guidelines for the elderly, such as Beers criteria, and they can be changed to another drugs.
We modified our manuscript as below; Although some of these factors are unpreventable, the influence of medication might be reduced by improving the quality of the prescription according to proper prescribing guidelines for the elderly, such as Beers criteria. (P2, line 67-70)
Point 4: Lines 57-58: “In addition, it is still unclear worldwide what proportion of falls in elderly care facilities are influenced by medication use” This is not a global study and hence authors may need to rephrase this to make it more relevant to their study.
Response 4: Thank you for comments. Our survey was conducted in four nursing homes, which account for about half of elderly care facilities in Japan. Moreover, the contents of care provide in these facilities were considered to be standard in Japanese nursing homes as far as chart review was concerned. However, the results in this study were only for the four facilities, and it is necessary to make further research to make an international comparison for generalization. We have added some description to the section of limitation about this topic. (page 10, line ●●)
Point 5: The authors claim that this is a historical study however they are focusing on the incidence and nature of falls. Authors need to clearly justify that such as study has not been conducted before if they are to call it “historical”.
Response 5: Thank you for your comment. We used “historical cohort study” as “retrospective cohort study”. We modified the word “historical” to “retrospective” in the title and the sections of abstract and method.
Point 6: • Figure 1 – include the number of people in your flow chart if possible, e.g. data was collected for how many patients? How many were excluded?
Response 6: Thank you for your comments. We excluded 4 falls because they were out of the definitions of fall. We modified figure 1.
Point 7: • How did the authors directly associate a fall as an adverse event of a drug? This needs to be more elaborate.
Response 7: Thank you for your comments. We associated a fall as adverse event of drug mainly by using Naranjo score, in addition to the previous method by Morimoto T “Adverse drug events and medication errors: detection and classification methods.” Quality Safety Health Care 2004, 13, 306–314.” And we modified some words of how to associate a fall as adverse event of drug as below; When the following situations were applicable, physicians evaluated whether each fall was an ADE and which medication most influenced the fall with calculating the Naranjo algorithm; medication would cause a fall due to its pharmacological action, medication was administered or discontinued within 24 hours of the occurrence of fall and if the administration or discontinuation of the medication could be the potential trigger of the fall. For the assessment of ADEs, the reviewers were trained according to a review manual based on the previous methods [26]. (p4, line 143-149)
We evaluated the degree of agreement between evaluators by using Cohen’s kappa, and we added about it on the section of result (p6, line 212-216).
Point 8: • The authors need to clearly identify and eliminate all other influencing factors that could have led to a fall apart from medicines. Once this is addressed in the methods, the results will be corroborated. Currently, the elimination of other factors is vague and therefore statements in results such as the proportion of people with medication-related falls and specific medicines need validation.
Response 8: Thank you for your suggestion. According to your indication, we added some words in the section of method as we mentioned in the response of #7; When the following situations were applicable, physicians evaluated whether each fall was an ADE and which medication most influenced the fall with calculating the Naranjo algorithm; medication would cause a fall due to its pharmacological action, medication was administered or discontinued within 24 hours of the occurrence of fall and if the administration or discontinuation of the medication could be the potential trigger of the fall. For the assessment of ADEs, the reviewers were trained according to a review manual based on the previous methods [26]. (p4, line 143-149)
Point 9: • “3.2. Comparison of the characteristics and risk factors between fallers and non-fallers among all 185 residents”. It would be interesting to discuss reasons why fallers had more medications, compared to non-fallers, despite having fewer morbidities.
Response 9: Thank you for your suggestion. As you mentioned, it would be interesting why falls had more medications despite having fewer morbidities. First, we assume that drugs for physical illness won’t have many factors on falls as the result of fall-caused drug in this study. Second, the number of complications may not be proportional to the number of drugs because some comorbidities need not take any medications by finishing or interrupting their treatments.
We added a few word in the section of discussion (p10 Line 298-300); Moreover, we may say that psychotropics would more influence for fall than physical medications because non-fallers had more comorbidities than fallers despite of higher of the number of comorbidities.
Point 10: • Lines 240-246: “high number of falls caused by antiepileptics may be a result of commissioned physicians’ attempt to avoid BZDRA and antipsychotics, which have been shown to be harmful to older adults with dementia…” • If authors are implicating that antiepileptics are used for other purposes (not for epilepsy), this should be discussed clearly and mentioned if this is according to standard practice guidelines in Japan.
Response 10: Thank you for your recommendation. Prescribing antiepileptics for BPSD are not strongly recommended in the guideline of treatment of dementia in Japan, however, many physicians used them in clinical settings. We added a few words in the manuscript.
Reviewer 2 Report
Thank you for giving me the chance to review this interesting manuscript. This study examined the association between falls and using psychotropic medications in nursing homes in Japan. Although the topic is of interest, there are some comments need to be addressed by the authors.
Abstract:
- You mentioned that “We also evaluated the relationship between potential risk factors for falls and injuries after falls.” However, it is very helpful if you mention these risk factors or at least highlight them in general such as balance measures or physical performance measures or chronic diseases …etc.
- The main outcome was fall. However, it is not clear how was collected and documented. It is important to specify the criteria for the main outcome.
- You mentioned here “Regularly taking ≥5 medications was a risk factor of falls (HR 1.33: CI 1.00–1.77, p=.0048) and injuries 24 after falls (OR 2.41: CI 1.30-4.50, p=.006)”, but it is need to be more specific here. For example, have you used receiver operator characteristics for number of medications to determine cutoff number? If not, I suggest using more appropriate analysis to answer this interesting question (what is the threshold for number of medications that are associated with falls?).
- You mentioned HR, and I am assuming you used cox proportional hazard. This should be explained before mentioning. Another concern for using survival analysis is the recurrent falls over time for the same subject that may affect the results.
- Your conclusion is overstated here “Continuous assessment to prevent polypharmacy and careful monitoring when prescribing psychotropic drugs are imperative in preventing falls in nursing homes.” I think it is too early to recommend these approaches using your data and results. This should be focused on the risk of fall and related risk factors such as medications.
Introduction:
- Well-written introduction related to the topic.
- The aim of the study needs some details such as determining the cutoff number of medications related to fall and the risk factors (what are they?)
Materials and methods:
- You mentioned details that are supposed to be in the discussion section such as “Since many residents in nursing homes are suffering from dementia [17] and they often take psychotropic drugs [18], psychiatrists and clinical psychologists as experts of behaviors and treatments of individuals with dementia took the lead in conducting this study under the supervision of two internists with sufficient experience in clinical epidemiological research..“ You can move these detail to the discussion part for reasoning and explanations.
- Definition: now it is the first time you mentioned the main outcome was number of falls. This should be clear from the abstract and purpose at the end of the introduction section.
- You still did not explain how falls were documented and defined. Self-reported? Nurse-documented? And timeframe.. over one week or month or daily. Please, specify the definition of falls and frequency of documenting falls because this will improve feasibility of replicating the study.
- The majority of psychotropic drugs can be used for pain relief. This raise pain severity or intensity as a covariate for the analysis.
- Please, use reference for classifying drugs into these classifications.
- Polypharmacy is a good definition. However, with older adults and living in nursing home, I think the cutoff 5 would be very common with more participants. Identifying a cutoff number is a good idea to be addressed here with your data.
- Data collection details is very nice, and I would suggest explaining falls again in the definition.
- Please, identify by names comorbidities to be clear to the audience.
- In the data analysis, you haven’t explained the analysis based on number of falls that you mentioned earlier. You used survival analysis for fall incidence but what about multiple incidences for the same participant?
- For examining number of falls, you can use Possion regression or negative binomial regressing with incidence rate ratio depending on overdispersion value.
- I would suggest using receiver operator characteristics and area under the curve to determine the cutoff value for some risk factors such as BI, number of medications, pain severity (if reported).
- You haven’t explained the analysis well. You mentioned multivariate (multivariable logistic) but you haven’t mentioned the adjusted model includes what or the controlled covariates what.
Discussion:
- I cannot evaluate at this moment until the methods and analysis become clear and answer the research question.
Author Response
Point 1: You mentioned that “We also evaluated the relationship between potential risk factors for falls and injuries after falls.” However, it is very helpful if you mention these risk factors or at least highlight them in general such as balance measures or physical performance measures or chronic diseases …etc.
Response 1: Thank you for your comments. We wrote to use Charlson Comorbidities index as evaluation of comorbidities, and that Barthel Index as evaluation of ADLs. We added some word in the abstract (p1, line 20-22) as below; We also evaluated the relationship between potential risk factors for falls and injuries after falls containing well-known risks such as ADLs and chronic comorbidities.
Point 2: The main outcome was fall. However, it is not clear how was collected and documented. It is important to specify the criteria for the main outcome.
Response 2: Thank you for your comment. We wrote in the section of method of data collection process as “The review process in this study was divided into three steps (Figure 1). First, five physicians and six clinical psychologists collaborated regarding the collection of the following data based on care records, prescriptions, laboratories, and incident reports” with figure 1. We added some words after that sentence; Each fall record in their nursing homes was written every day by observation of nurses and staffs on their care charts and incident reports. (p3, line 146-148).
Point 3: You mentioned here “Regularly taking ≥5 medications was a risk factor of falls (HR 1.33: CI 1.00–1.77, p=.0048) and injuries 24 after falls (OR 2.41: CI 1.30-4.50, p=.006)”, but it is need to be more specific here. For example, have you used receiver operator characteristics for number of medications to determine cutoff number? If not, I suggest using more appropriate analysis to answer this interesting question (what is the threshold for number of medications that are associated with falls?).
Point 6: The aim of the study needs some details such as determining the cutoff number of medications related to fall and the risk factors (what are they?)
Point 12: Polypharmacy is a good definition. However, with older adults and living in nursing home, I think the cutoff 5 would be very common with more participants. Identifying a cutoff number is a good idea to be addressed here with your data.
Response 3, 6, and 12: We appreciate for your significant recommendation. We would like you to allow us to answer these three questions at once. On this study, we estimated the number of falls in this survey from the general fall incidence, and set the number of variables to be included in the multivariate analysis to about seven. After that, we selected risks of falls and injuries of falls as sex, age, medical history and comorbidities, presence of dementia, functional independence and the number of prescriptions, which were referred in the previous studies and deeply associated we judged by the clinical assessment. As you said, we found there were many definitions of polypharmacy in the previous studies of falls as below.
Four or more: Ziere 2006 (PMID: 16433876 PMCID: PMC1885000 DOI: 10.1111/j.1365-2125.2005.02543.x), Anam Zia 2015 (PMID: 25539567 DOI: 10.1080/00325481.2014.996112), Nafeesea 2017(PMID: 29042378, DOI: 10.1136/bmjopen-2017-016358)
Five or more:Masumoto 2018 (PMID: 29582533 DOI: 10.1111 / ggi.13307), Kojima 2011 (PMID: 22212467 DOI: 10.1111/j.1447-0594.2011.00783.x), Anam Zia 2016 (PMID: 26822931, DOI: 10.1111 / ggi.12741), Hamza 2019 (PMID: 30920085 DOI: 10.1002/pds.4775)
Seven or more: Federico 2009 (PMID: 20003327, DOI: https://doi.org/10.1186/1472-6963-9-228)
Considering these studies, we decided our definition of fall referred to Masnoon 2017 (What is polypharmacy? A systematic review of definitions. BMC Geriatr. 2017;17:230.).
We didn’t use the ROC to decide out cut-off because we would like to survey the association between polypharmacy, falls and injuries after falls by using the consensus of definition in the same research field, not to identify the criteria of polypharmacy of our survey.
Point 4: You mentioned HR, and I am assuming you used cox proportional hazard. This should be explained before mentioning. Another concern for using survival analysis is the recurrent falls over time for the same subject that may affect the results.
Response 4: Thank you for your comments. We use multivariate Cox proportional hazard models to compare the associations of potential risk factors for first falls as the occurrence of falls, adjusting for differences in length of study during the survey period. To examine this method, we modified the description of abstract and the section of data analysis as below; Multivariate Cox proportional hazards models were used to compare the association of potential risk factors of first falls as the occurrence of falls, adjusting for differences in the length of residential stay during the study period. (p5, line 188-190).
Point 5: Your conclusion is overstated here “Continuous assessment to prevent polypharmacy and careful monitoring when prescribing psychotropic drugs are imperative in preventing falls in nursing homes.” I think it is too early to recommend these approaches using your data and results. This should be focused on the risk of fall and related risk factors such as medications.
Response 5: Thank you for your comments. Based on your suggestion, we modified our words in the abstract as below to make our message clearly; Preventing the occurrence and persistence of polypharmacy by discontinuing unnecessary medications may lead avoid falls in nursing homes while consulting with physicians and pharmacists after adequate assessment of ADLs and comorbidities (page 1, line 28-30).
Point 7: You mentioned details that are supposed to be in the discussion section such as “Since many residents in nursing homes are suffering from dementia [17] and they often take psychotropic drugs [18], psychiatrists and clinical psychologists as experts of behaviors and treatments of individuals with dementia took the lead in conducting this study under the supervision of two internists with sufficient experience in clinical epidemiological research..“ You can move these detail to the discussion part for reasoning and explanations.
Response 7 Thank you for your comments. As you mentioned, the sentence in the section of introduction were suitable for in the section of discussion. We thought this is our strong point of study, we moved this sentence in the section of discussion in p9 line 291-295.
Point 8: Definition: now it is the first time you mentioned the main outcome was number of falls. This should be clear from the abstract and purpose at the end of the introduction section.
Response 8: Thank you for comments. We described our purpose of this study in the ind of the introduction section as “we planned to identify ” we planned to identify the incidence and nature of falls with and without injury at four nursing homes in Japan, the most rapidly aging society in the world.”. However we didn’t write down our purpose in the abstract. Therefore, we added a few words on the abstract (p1, line18-20).
Point 9: You still did not explain how falls were documented and defined. Self-reported? Nurse-documented? And timeframe.. over one week or month or daily. Please, specify the definition of falls and frequency of documenting falls because this will improve feasibility of replicating the study
Response 9: Thank you for your comment. We wrote in the section of method of data collection process as “The review process in this study was divided into three steps (Figure 1). First, five physicians and six clinical psychologists collaborated regarding the collection of the following data based on care records, prescriptions, laboratories, and incident reports” with figure 1 as previously stated. We added some words after the sentence to improve and clarify our explanation; Each fall record in their nursing homes was written every day by observation of nurses and staffs on their care charts and incident reports. (p3, line 146-148).
Point 10: The majority of psychotropic drugs can be used for pain relief. This raise pain severity or intensity as a covariate for the analysis.
Point 11: Please, use reference for classifying drugs into these classifications.
Response 10 and 11: We appreciate your comment that we should consider psychotropic drugs can be used for pain relief. As you said, some psychotropics are prescribed as to relieve pain. However, because these kinds of drugs were elementally used for psychiatric symptoms even if they were used for pain relief, they were classified as a psychotropic drug. To apply this classification on this study, we referred previous studies by Morimoto 2010 (doi: 10.1007/s11606-010-1518-3, PMID: 20872082), Sakuma 2014(PMID: 24742779 DOI: 10.1136/bmjqs-2013-002658), and Ayani 2016(doi: 10.1186/s12888-016-1009-0, PMID: 27577925).
In fact, looking at individual data, there weren’t many factors used for pain. Those that can also be used for pain relief are classified as psychotropic drugs.
Point 13: Data collection details is very nice, and I would suggest explaining falls again in the definition.
Response 13: Thank you for your comments. As your recommendation, we added some words as below; Descriptions of which, for example, residents fell after some trouble with other residents were excluded.
Point 14: Please, identify by names comorbidities to be clear to the audience.
Response 14: Thank you for your suggestion. As you said, we would like to write each comorbidity, but we have limitations of manuscript volume and we used Charlson Comorbidities Index.
Point 15: In the data analysis, you haven’t explained the analysis based on number of falls that you mentioned earlier. You used survival analysis for fall incidence but what about multiple incidences for the same participant? For examining number of falls, you can use Possion regression or negative binomial regressing with incidence rate ratio depending on overdispersion value.
Response 15: Thank you for your comment. As you said, examining number of falls is important sight, however, we could not survey numbers of falls of residents who had stayed before we started our study. Then to adjust for diffences in length of study during the survey period, We use multivariate Cox proportional hazard models to compare the associations of potential risk factors for first falls as the occurrence of falls. To examine this method, we modified our explanation in the section of data analysis as below; Multivariate Cox proportional hazards models were used to compare the association of potential risk factors of first falls as the occurrence of falls, adjusting for differences in the length of residential stay during the study period. (p5, line 188-190)
Point 15-2: I would suggest using receiver operator characteristics and area under the curve to determine the cutoff value for some risk factors such as BI, number of medications, pain severity (if reported).
Response 15-2: We appreciate for your suggestion. As we said in the response 3, we estimated the number of falls in this survey from the general fall incidence, and after that we set the number of variables to be included in the multivariate analysis by being referred to previous studies. Through this process, we selected risks of falls and injuries of falls to use in analysis.
We investigated the relationship between the Barthel Index and the number of medications for residents exposure to a fall more than once and found that residents with low levels of ADLs took more medications. Then we assumed that residents of low level of ADLs lead to more falls from closer to the groud, such as beds and wheelchairs, and less likely to result in trauma.
Point 16: You haven’t explained the analysis well. You mentioned multivariate (multivariable logistic) but you haven’t mentioned the adjusted model includes what or the controlled covariates what.
Response 16: We appreciate your comments. In order to make our message clear, we modified our manuscript to the sections you pointed out on p5 Line 163-164; The potential risk factors in both adjusted models, the multivariate Cox proportional hazards model and the multivariate logistic regression model, included sex, older age (≥85 years), presence of dementia, number of medications at the start of the study, CCI (≥3), and BI (independent and mild-dependent (≥60).
Point 17: I cannot evaluate at this moment until the methods and analysis become clear and answer the research question.
Response 17: We would highly like you to evaluate our modified manuscript and make comments and recommendations
Reviewer 3 Report
This retrospective record review offers useful insights into an important problem.
Tables
- Avoid isolated P values: test results + df should be stated.
- Where 95% confidence intervals are quoted, P values are superfluous.
How was patient mobility accounted? In any long-term care facility the most mobile patients are those with mental health problems, prescribed antipsychotics: patients with a physical incapacity are often bedridden.
AEDs were associated with falls: were these due to seizures?
Data relate to 2016-2017. Are more recent data available? How do these findings relate to current, post-covid, practice?
Generalisability beyond Japan should be discussed: are prescribing patterns similar?
If medications at the time of the fall are unknown (line 268 et seq), the value of the data is questionable. Since the dates of the falls and prescribing information will be recorded, this can be rectified.
Are medication doses available?
Careful editing by a native English speaker with subject knowledge is needed: some sentences are difficult to understand e.g. lines 255 et seq.
Author Response
Point 1 : Avoid isolated P values: test results + df should be stated.
Point 2: Where 95% confidence intervals are quoted, P values are superfluous.
Response 1 and 2: Thank you for your comments. We know the recent trends of writing tables without p-value in the field of statistics. On the other hand, we think that many readers are still helped to understand easily by the notation of p-value. Thus, we deleted p-value from table 3 (Hazard ratios of fall occurrence from Cox proportional hazard models) and table 5 (Odds ratios of injury occurrence after falls based on logistic regression analysis) and we remain to write p-values on table 1 (Characteristics of residents.) and table 4 (Characteristics between fallers with and without injuries.). Regarding to degree of freedom, it can regarded as equal to the normal distribution, and we consider it could be omitted.
Point 3: How was patient mobility accounted? In any long-term care facility the most mobile patients are those with mental health problems, prescribed antipsychotics: patients with a physical incapacity are often bedridden.
Response 3: Thank you for your comments. I agreed with your indicate. We recorded mobility of each resident, and we analyzed the ADL as Barthel Index, including items of transfer (bed to chair and back) and mobility on level surface. We analyzed the association between the prescription of antipsychotics and the level of Barthel Index (in 2 levels and in 4 levels) , but there were no association in our study.
Point 4: AEDs were associated with falls: were these due to seizures?
Response 4: Thank you for your comments. As you mentioned, antiepileptic drugs were associated with falls, however, these falls didn’t associate with seizures because the definition of falls excluded seizures, as we said in Line 81-84.
Point 5: Data relate to 2016-2017. Are more recent data available? How do these findings relate to current, post-covid, practice?
Response 5: Thank you for your suggestion. I’m sorry we don’t have recent data, however, we would like to survey this theme in future.
Point 6: Generalisability beyond Japan should be discussed: are prescribing patterns similar?
Response 6: Thank you for your comment. As you said, it may not clear whether this study can be generalized to the current situation in Japan, however, these four nursing homes are very ordinary facilities because these types of nursing homes are more than half of the total in Japan. Moreover, we judged the cares clinical settings in these four nursing homes were ordinary as reading all of care records and as visiting them during the period of study.
Thus, we think it can be inferred that similar results would be obtained even if the same research is conducted at multiple facilities. Of course, it is necessary to make an international comparison for generalization.
Point 7: If medications at the time of the fall are unknown (line 268 et seq), the value of the data is questionable. Since the dates of the falls and prescribing information will be recorded, this can be rectified.
Response 7: Thank you for your comments. In analysis of risk factors of falls and injuries, we used prescriptions at the start of the survey as characteristic information on the residents. On the other hand, we identified specific medications influenced on falls by whole chart review. We used established methodologies to evaluate risks of ADEs by using prescriptions at the star of the survey though they might be different from those when the events occurred.
Point 8: Are medication doses available?
Response 8: Thank you for comments. We evaluated by using Naranjo score. Furthermore, we discussed the associations between circumstances of fall occurrence and dosage of medications. We calculated and added the kappa score in the manuscript. (Line 187-190).
Point 9: Careful editing by a native English speaker with subject knowledge is needed: some sentences are difficult to understand e.g. lines 255 et seq.
Response 9: Thank you for your comments. We have requested another proofreading in English.
Round 2
Reviewer 1 Report
Thank you for addressing all comments. The methods appear to be clearer now.
Author Response
Point 1: Thank you for addressing all comments. The methods appear to be clearer now.
Response 1: Thank you for yoru suggestions. As a result of following your advices, we were able to improve our manuscript much better.
Reviewer 2 Report
The authors have revised and justify based on my comments
Author Response
Point 1: The authors have revised and justify based on my comments
Response 1: Thank you for your suggestions. As a result of following your advices, we were able to improve our manuscript much better.
Reviewer 3 Report
Some of the points raised have been addressed.
The absence of medicines at the time of falls and doses are major limitations that should be discussed in full.
I agree with the conclusion that more monitoring for adverse drug reactions is needed. The authors may be interested in how this can be achieved to reduce falls in this population [1], and the wider issues [2].
It is the editors’ decision as to whether isolated P values are acceptable: in my view they are not.
English language editing by a native speaker with subject knowledge is essential. For example, ‘incidence’ is substituted for ‘prevalence’.
I hope I shall receive a pdf on publication.
- Jordan S, Prout H, Carter N, Dicomidis J, Hayes J, Round J, Carson-Stevens A. (2021) Nobody ever questions—Polypharmacy in care homes: A mixed methods evaluation of a multidisciplinary medicines optimisation initiative. PLOS ONE 16(1): e0244519. https://doi.org/10.1371/journal.pone.0244519
- Jordan,S., Logan,V., Turner,A., & Hughes,D.Using nurse-led patient monitoring to avoid medicines-related harm.Nursing Standard, doi:10.7748/ns.2021.e11770. 28.6.21 Using nurse-led patient monitoring to avoid medicines-related harm (rcni.com) Open access https://journals.rcni.com/nursing-standard/evidence-and-practice/using-nurseled-patient-monitoring-to-avoid-medicinesrelated-harm-ns.2021.e11770/full
Author Response
Some of the points raised have been addressed.
The absence of medicines at the time of falls and doses are major limitations that should be discussed in full.
Thank you for your comment. We agreed that The absence of medicines at the time of falls and doses are major limitations that should be discussed. We apologized for our poor expression that we didn’t analyze with the medication factor at the time of falls. We had written, “Multivariate analysis was used to assess the association of drugs as a risk factor for falls and fall-related injuries, but only included the number of medications taken at the start of the survey.” Therefore, we added some words to improve our manuscript as below (p10 lines 306-309); In the strict sense, there were some differences in prescriptions between those at the start of the survey and those at the time of falls. However, previous studies also used the factors at the beginning of the survey as representative values for each resident in the analysis.
I agree with the conclusion that more monitoring for adverse drug reactions is needed. The authors may be interested in how this can be achieved to reduce falls in this population [1], and the wider issues [2].
Thank you for your comments with suggestive papers and understanding our opinion that more monitoring for adverse drug reactions was needed. As you mentioned, educations and interventions for medical staffs are also important for reducing ADEs of residents in nursing homes. We added some words with your suggested papers as below (p9, line 533-535): Therefore, doctors themselves should be careful about their prescribing and monitoring and nurses and carers would be educated on medications and interventional programs.
It is the editors’ decision as to whether isolated P values are acceptable: in my view they are not.
Thank you for your comments. For you recommendation, we added χ2 or t-value on the table 1 and table 4 referred to a previous paper [1]. We hope our correcitons will meet the requirements of yours and the editors.
- Davis AK, Barrett FS, May DG, et al. Effects of Psilocybin-Assisted Therapy on Major Depressive Disorder: A Randomized Clinical Trial [published correction appears in JAMA Psychiatry. 2021 Feb 10;:]. JAMA Psychiatry. 2021;78(5):481-489. doi:10.1001/jamapsychiatry.2020.3285
English language editing by a native speaker with subject knowledge is essential. For example, ‘incidence’ is substituted for ‘prevalence’.
Thank you for your comments. We received English language editing by a native speaker again, and at the same time we recognized the definition of the word “incidence” and “prevalence” as below.
Incidence: new cases that occurred during a given time
Prevalence: all cases present during a given time period
(Referred; https://www.cdc.gov/csels/dsepd/ss1978/lesson3/section2.html#:~:text=Prevalence%20refers%20to%20proportion%20of,during%20a%20particular%20time%20period)
We calculated how many and how often falls occurred during only our survey period, and there were general to use the word incidence in similar situations of previous studies[2-4]. Then we used the word incidence on our study.
- Morimoto, T. Adverse drug events and medication errors: detection and classification methods. Quality Safety Health Care 2004, 13, 306–314.
- Tanaka, B.; Sakuma, M.; Ohtani, M.; Toshiro, J.; Matsumura, T.; Morimoto, T. Incidence and risk factors of hospital falls on long-term care wards in Japan. J Eval Clin Pract 2012, 18, 572–577.
- Ayani, N.; Sakuma, M.; Morimoto, T.; Kikuchi, T.; Watanabe, K.; Narumoto, J.; Fukui, K. The epidemiology of adverse drug events and medication errors among psychiatric inpatients in Japan: the JADE study. BMC Psychiatr 2016, 16, 303.
I hope I shall receive a pdf on publication.
- Jordan S, Prout H, Carter N, Dicomidis J, Hayes J, Round J, Carson-Stevens A. (2021) Nobody ever questions—Polypharmacy in care homes: A mixed methods evaluation of a multidisciplinary medicines optimisation initiative. PLOS ONE 16(1): e0244519. https://doi.org/10.1371/journal.pone.0244519
- Jordan,S., Logan,V., Turner,A., & Hughes,D.Using nurse-led patient monitoring to avoid medicines-related harm.Nursing Standard, doi:10.7748/ns.2021.e11770. 28.6.21 Using nurse-led patient monitoring to avoid medicines-related harm (rcni.com) Open access https://journals.rcni.com/nursing-standard/evidence-and-practice/using-nurseled-patient-monitoring-to-avoid-medicinesrelated-harm-ns.2021.e11770/full